# Identification of Three Viruses Infecting Mulberry Varieties

**DOI:** 10.3390/v14112564

**Published:** 2022-11-19

**Authors:** Lei Chen, Zi-Long Xu, Pei-Gang Liu, Yan Zhu, Tian-Bao Lin, Tian-Yan Li, Zhi-Qiang Lv, Jia Wei

**Affiliations:** 1Institute of Sericulture and Tea, Zhejiang Academy of Agricultural Sciences, Hangzhou 310000, China; 2College of Horticulture Science, Zhejiang A&F University, Hangzhou 310000, China

**Keywords:** virus, mulberry alba, transcriptome, *Citrivirus*, *Virgaviridae*, *Narnavirus*

## Abstract

Viruses-mediated genome editing in plants is a powerful strategy to develop plant cultivars with important and novel agricultural traits. Mulberry alba is an important economic tree species that has been cultivated in China for more than 5000 years. So far, only a few viruses have been identified from mulberry trees, and their application potential is largely unknown. Therefore, mining more virus resources from the mulberry tree can pave the way for the establishment of useful engineering tools. In this study, eight old mulberry plants were gathered in seven geographic areas for virome analysis. Based on transcriptome analysis, we discovered three viruses associated with mulberries: Citrus leaf blotch virus isolate mulberry alba 2 (CLBV-ML2), Mulberry-associated virga-like virus (MaVLV), and Mulberry-associated narna-like virus (MaNLV). The genome of CLBV-ML2 was completely sequenced and exhibited high homology with Citriviruses, considered to be members of the genus *Citrivirus*, while the genomes of MaVLV and MaNLV were nearly completed lacking the 5′ and 3′ termini sequences. We tentatively consider MaVLV to be members of the family *Virgaviridae* and MaNLV to be members of the genus *Narnavirus* based on the results of phylogenetic trees. The infection experiments showed that CLBV-ML2 could be detected in the inoculated seedlings of both *N. benthamiana* and *Morus alba*, while MaVLV could only be detected in *N. benthamiana*. All of the infected seedlings did not show obvious symptoms.

## 1. Introduction

Next-generation sequencing (NGS) serves as an unbiased technology for the diagnosis of plant viral diseases since no prior information about the pathogen is required [1]. Resent day NGS tools are capable of sequencing any type of nucleic acid and have emerged as the tool of choice to detect novel viral diseases from very few viral titers [2]. Many novel viruses were identified by NGS, which before were quite difficult because of either low viral titer or difficulty of RNA extractions from woody plants or the absence of any symptoms. Recently, transcriptome sequencing was used in citrus to identify a novel latent viral infection, *citrus yellow vein-associated virus* (CYVaV), and analysis showed that it could be an ancestral virus to the current umbravirus [3,4].

The mulberry (*Morus alba*) belongs to the genus *Morus* in the family *Moraceae*, which is a deciduous tree that has been grown for thousands of years in China [5] The leaves of mulberry trees are the only food source for silkworms [6] and also can be converted into mulberry tea [7] and other beverages [8]. Their fruits are rich in nutrients and can be used for preparing fruit juice and wine [9,10]. The yield and quality of mulberry leaves can be seriously affected by a virus infection, causing severe loss to the sericultural industry. Before the application of NGS, only a few viruses had been identified in mulberry trees, such as the *Mulberry crinkle leaf virus* (MCLV) [11] and the *Paper mulberry mosaic-associated virus* (PMuMaV) [12]. With the rapid development of NGS technology, much more viruses have been identified in trees of mulberry recently. *Mulberry mosaic dwarf-associated virus* (MMDaV) caused mosaic and dwarfing symptoms in mulberry trees [13], and *mulberry cryptic virus 1* (MuCV1) exhibited yellow vein symptoms [14]. The symptoms caused by *mulberry badnavirus 1* (MBV-1) include mosaic structure, deformation, vein clearing and necrosis on the leaves and deformation, crumbling, and scabs on the fruits [15]. *Citrus leaf blotch virus-ML* (CLBV-ML) caused symptoms of chlorotic leaf spots and occasionally witches’ broom [16]. In addition, *paper mulberry leaf curl virus 1* and *paper mulberry leaf curl virus 2* (PMLCV-1 and PMLCV-2) were identified in paper mulberry plants affected by a disease with leaf curl symptoms [17]. However, there are still many viruses that have not been identified yet, especially those with either low accumulation or weak symptoms in mulberry trees. Identification of novel viruses in mulberry is meaningful.

Nowadays, plenty of old mulberry trees, which have lived for hundreds and thousands of years, are mainly distributed in the middle and lower reaches of the Yellow River, southwest, northwest, and northeast of China. In this study, Transcriptome sequencing technology was applied to eight old mulberry trees’ leaves from seven provinces in order to uncover novel viruses that co-existences with their hosts. Three viruses were detected from the sequence data, one of which is *citrus leaf blotch virus* (CLBV), a member of the genus *Citrivirus*. We named it Citrus leaf blotch virus-Mulberry Alba 2 (CLBV-ML2) since it is homologous to the previously reported mulberry virus *Citrus leaf blotch virus-ML* [16]. The other two are new viruses belonging to the *Virgaviridae* and *Narnavirus*, with the names of Mulberry-associated virga-like virus (MaVLV) and Mulberry-associated Narna-Like virus (MaNLV) proposed. RT-PCR experiments have also been conducted with specific RdRp primers of these viruses to verify their existence. Besides, the results of virus infection experiments showed that MaVLV could infect *N. benthamiana*, and CLBV-ML2 could be detected in the inoculated seedlings of both *N. benthamiana* and *Morus alba* without obvious symptoms.

## 2. Materials and Methods

### 2.1. Plant Material 

The present study examined eight different species of ancient mulberry trees from Chinese provinces (Heilongjiang, Henan, Shandong, Anhui, Beijing, Xinjiang, and Xizang). The samples were leaves of ancient mulberry trees, which were asymptomatic at the time of collection. They were stored in sterile 50 mL conical tubes at −80 °C until further analysis.

### 2.2. Sequence Analyses

The obtained samples were sent to Majorbio (Shanghai, China) for total RNA isolation and transcriptome sequencing and analysis. Raw data were trimmed using Trimmomatic (Version 3.90) [18] and trinity (Version 2.8.5) with the default parameters used for contig assembly [19]. Then, blastx from the BLAST suite was used to search for conserved viral proteins in the assembled contigs [20]. Finally, open reading frames (ORFs) were predicted using the ORFfinder web tool (https://www.ncbi.nlm.nih.gov/orffinder/, accessed on 15 January 2022). A conserved domain search was performed using the CDD/SPARCLE tool (https://www.ncbi.nlm.nih.gov/Structure/cdd/wrpsb.cgi/, accessed on 15 January 2022) of NCBI [21]. TBtools v0.67 software was used to display the pairwise identity of the sequences aligned by the ClustalW program and prepare a correlation heatmap.

### 2.3. Phylogenetic Analyses

The amino acid sequences of the predicted RNA-dependent RNA polymerase (RdRp) regions from MaVLV and MaNLV were used for phylogenetic analyses in this study. The sequences were obtained from National Center for Biotechnology Information (NCBI, https://www.ncbi.nlm.nih.gov/, accessed on 15 January 2022), compared with the related viral sequences using ClustalW and analyzed using the Maximum likelihood (ML) algorithm with the LG and WAG substitution model to construct phylogenetic trees in MEGA X with 1000 bootstrap replications, respectively [22]. The whole sequence of CLBV-ML2 was used for phylogenetic analyses. The sequences obtained from NCBI and subject to ClustalW multiple sequence alignments. Then, the maximum likelihood method based on the TN93 matrix-based models with 1000 bootstrap replications was separately conducted.

### 2.4. Completion of Genomes of Viruses

The full genome sequence of the candidate CLBV-ML2 was obtained by rapid amplification of cDNA ends (RACE) using a SMARTer^®^ RACE 5′/3′ kit (Takara, Beijing, China). The 5′ RACE CDS Primer A and the 3′ RACE CDS Primer A was used to synthesize the cDNAs, which could include specific amplification of 5′ and 3′ end sequences, respectively. The universal primer mix and the gene-specific primers (GSP, Appendix A) were used to complete the 5′-terminal and 3′-terminal sequences with 2×Hieff Canace^®^ Plus PCR Master Mix (Yeasen, Shanghai, China). The purified PCR product was cloned into a pESI-T vector (Yeasen, Shanghai, China), and the M13-F/R primer was used for positive clone detection and sequenced commercially (Youkang, Hangzhou, China). Finally, the whole sequence was assembled with DNAMAN software (version 9.0), and observed that an additional 66 nt at the 5′ terminus and 528 nt at the 3′ terminus was identified.

### 2.5. Mechanical Inoculation and Virus Detection by RT-PCR

Mechanical inoculation was used for virus infection. The basic procedure was to grind up the samples from ancient mulberry trees and add the appropriate amount of phosphate buffer (250 mL 0.2 mol/L KH_2_PO_4_, 118 mL 0.2 mol/L NaOH, add H_2_O to 1000 mL, pH 6.8), then rub the extracted sap onto the leaves of 5 *N. benthamiana* and 5 *Morus alba*. Total RNA was extracted from system leaves after 10 days. RT-PCR was used for the RdRp sequence detection from the samples’ total RNA and inoculated plants’ total RNA. Following the manufacturer’s instructions, the RT reaction was performed with Hifair^®^ III 1st Strand cDNA Synthesis Kit (Yeasen, Shanghai, China) with Oligo (dT)_18_ primer. One microliter of cDNA was used as a template for the PCR reaction in the presence of primers of RdRp detection. The reaction procedure was 94 °C for 3 min; 94 °C for 30 s; 55 °C for 30 s; 72 °C for 1 min (30 s/kb); 72 °C for 5 min, and end of 4 °C, which produced 1295 bp, 729 bp, and 1298 bp fragments, respectively.

## 3. Results

### 3.1. Discovery of Viruses and Sequencing Results

The present virome analysis was performed on the transcriptome sequencing samples of eight libraries from mulberry leaves, which led to the discovery of three viruses (Table 1). The heatmap shows the distribution of MaVLV, CLBV-ML2, and MaNLV in the transcriptome sequences (Figure 1).

The virus, MaVLV, was a contig of 11207 nt, excluding the sequences of the 5′ and 3′ UTRs. Its sequence shares 72.76% and 73.68% nt similarity with the Hubei virga-like virus 18 (HVLV18) strain fly97447 (KX883767) and Broome virga-like virus 1 (BVLV1) isolate BVGL1/pool-2 (MT498833), which are unclassified virgaviruses found in invertebrates, respectively. Four ORFs were found using the ORFfinder web tool. ORF1 is positioned at 98–7372 nt in the genome, predicted to encode a 2424 aa polyprotein including a viral methyltransferase (MT, pfam01660, nt 371–1411, 365 aa), an FtsJ-like methyltransferase (FtsJ, pfam01728, nt 2627–3148, 174 aa), a viral RNA helicase (HEL, pfam01443, nt 4826–5606, 260 aa), and an RNA-dependent RNA polymerase 2 domain (RdRp_2, pfam00978, nt 6023–7318, 432 aa). ORF2 is positioned at 8256–9257 nt in the genome, which encodes unknown proteins. ORF3 and ORF4 are positioned at 9291–10061 nt and 10081–10614 nt in the genome, predicted to encode 154 and 151 aa SP24, respectively (Figure 2A).

CLBV-ML2 has an 8794 nt genome with 66 nt UTR at its 5′ terminus and 528 nt UTR at its 3′ terminus. The 5′-UTR with a length of 66 nt shares 98.48% nt identity with that of CLBV-PY2 (MW713063), and the 3′-UTR with a length of 528 nt shares 95.83% nt identity with CLBV-ML (MT767171). The complete genomic sequence shares 83.08% and 81.87% identity with CLBV-PY2 and CLBV-ML, respectively. Similar to other citriviruses, CLBV-ML2 has three ORFs. ORF1 is positioned at 67–6024 nt in the genome and encodes a 1986 aa polyprotein, including a viral methyltransferase (MT, pfam01660, nt 196–1008, 271 aa); a 2OG-FeII oxygenase (2OG-F, pfam13532, nt 2669–3016, 116 aa); a peptidase (Pep, pfam05379, nt 3071–3331, 87 aa), and an RNA-dependent RNA polymerase 2 domain (RdRp_2, pfam00978, nt 5015–5743, 243 aa). ORF2 is positioned at 6022–7110 nt in the genome, and ORF3 is positioned at 7171–8264 nt in the genome, encoding a movement protein and a coat protein, respectively (Figure 2B). 

MaNLV has a 4097 nt genome, excluding the 5′ and 3′ UTR sequences. Its sequence shares 79.21% nt similarity with the *Narnaviridae* sp. isolate 174-k141_105200 (MZ680060), which was found in pond sediment. Two ORFs were found by the ORFfinder web tool. ORF1 is positioned at 73–789 nt in the genome, and ORF2 is positioned at 725–3991 nt in the genome and possibly encodes RdRp protein. However, no conserved domain was found by the CDD/SPARCLE search (Figure 2C).

### 3.2. Phylogenetic Placement of Identified Viruses

The phylogenetic tree is based on the RNA-dependent RNA polymerase (RdRp) amino acid sequences of representative members of the *Virgaviridae* family. The phylogenetic analysis result shows that MaVLV falls in the clade with Grapevine-associated virga-like virus (QXN75450) and belongs to the unclassified *Virgaviridae* (Figure 3A). We propose that MaVLV was a member of unclassified virgaviridaes because its RdRp amino acid sequence showed low similarity with isolates of GaVLV in the correlation heatmap (Figure 3B).

The phylogenetic trees based on the whole sequences of representative members of the *Citrivirus* genus show that CLBV-ML2 identified in the present study falls in the clade where sequences of CLBV-PY2 and -PY1 (Figure 4A). The correlation heatmap shows that the whole sequence of CLBV-ML2 is conserved among citriviruses (Figure 4B). The nucleotide and putative amino acid sequences of CLBV-ML2 were compared with those of the CLBV isolates and unassigned viruses. The whole genome of CLBV-ML2 showed nucleotide identity higher than 75% (Table 2). At the nucleotide sequence level, each ORFs of CLBV-ML2 showed nucleotide identities ranging from 53.35 to 88.81%, 58.07 to 98.07%, and 78.51 to 95.32%. At the amino acid sequence level, they showed similar sequence identities ranging from 75.44 to 82.44%, 79.76 to 84.02%, and 83.12 to 84.59% (Table 2). Based on the species demarcation criteria for *Betafexiviridae* [23], we propose that CLBV-ML2 is a new isolate of *Citrivirus* because it has similar polyprotein processing programs and share more than 70% amino acid identity in the polyprotein with any other citriviruses.

The phylogenetic tree is based on the RdRp amino acid sequence of representative members of the *Narnaviridae* family. The phylogenetic analysis result shows that MaNLV falls in the clade with *Neofusicoccum parvum narnavirus*, which was found in grapevine wood tissues and is a member of unclassified narnaviruses (Figure 5A). Based on the correlation heatmap, we considered that MaNLV belongs to the genus *Narnavirus* (Figure 5B).

### 3.3. The Detection of Viruses from Samples and Complete Sequence of CLBV-ML2

To further confirm the transcriptome sequencing and analysis results and identify the complete virus genomic sequence. Total RNA was extracted from the diseased leaves for the subsequent PCR and RACE experiments. The results of RT-PCR showed that the RdRp of MaVLV and CLBV-ML2 can be detected, but the passage of MaNLV was negative. Therefore, we determined that MaVLV and CLBV-ML2 were latent in G026 and G039 samples (Figure 6A). The 5′-terminal and 3′-terminal sequences of CLBV-ML2 were amplified by the RACE kit (Figure 6B). Then, the purification of PCR products was cloned into the pESI-T vector to screen positive clones and sequenced commercially (Appendix A). Finally, the 66 nt 5′-UTR (1–66 nt) and 528 nt 3′-UTR (8265–8792 nt) were obtained.

### 3.4. Viruses Infectivity in N. benthamiana and Morus alba

In order to investigate the infectivity of this novel virus, mechanical inoculation was used for virus infection by grinding up the samples from mulberry trees and adding an appropriate amount of phosphate buffer. We then rubbed the extracted sap onto the leaves of five *Nicotiana benthamiana* and five *Morus alba*. Ten days later, there were no symptoms on the upper leaves of the inoculated seedlings (Figure 7A and Appendix A). Total RNA was extracted from the infection leaves. RT-PCRs with specific primers of RdRp (Appendix A) were performed against the part infection leaves. There were weak stripes for MaVLV RdRp sequences in the cDNA from *N. benthamiana* and nothing in mock or cDNA from *Morus alba* compared to the result of the G026 sample with the mock and infection. Both *N. benthamiana* and *Morus alba* lanes had obvious stripes in the detection of CLBV-ML RdRp sequences, and their sizes were consistent with the sample detection results (Figure 7B). In addition, the test results for MaNLV infectivity were negative (Appendix A). We concluded that of the possible reasons for this phenomenon, one possibility is viral titers in collected mulberry were low, and few plants were inoculated. The other possibility is inoculation buffer is inappropriate, especially the pH-value.

## 4. Discussion

This study uncovered three viruses connected to mulberry based on transcriptome sequencing of eight mulberry samples. A full-length sequence of CLBV-ML2, which exhibited high homology and clustered with citriviruses in phylogenetic trees. The other two viruses, MaVLV and MaNLV, belong to the member of unclassified virgaviridaes and unclassified narnaviruses by genome and phylogenetic analysis.

Virgaviridae family consists of plant viruses with rod-shaped virions, a single-stranded RNA genome with a 3′-terminal tRNA-like structure, and an alpha-like replication protein. Its member genera include *Furovirus*, *Hordeivirus*, *Pecluvirus*, *Pomovirus*, *Tobamovirus*, and *Tobravirus* [24,25]. In this study, we identified a novel virus, MaVLV, in mulberry plants, which was presumed to be a member of virgaviruses. Interestingly, similarly to the member of virgaviridae, MaVLV ORF1 is also an alpha-like replication protein with conserved methyltransferase (MT) and helicase (Hel) domains. The RNA-dependent RNA polymerase (RdRp) is expressed as the C-terminal part of this protein by the readthrough of a leaky stop codon. But the movement protein and coat protein have not been proven to encode in the MaVLV genome. Due to the lack of 5’UTR and 3’UTR sequences in the genome, the existence of a 3’-terminal tRNA-like structure needs future investigation.

Narnaviruses have been described as positive-sense RNA viruses with a remarkably simple genome of ~3 kb, encoding only a highly conserved RNA-dependent RNA polymerase (RdRp) [26]. They could be detected in diverse fungi, plants, protists, arthropods and nematodes [27]. In this study, we identified a novel virus MaNLV with uncomplete genome sequence of about 4097 nt in mulberry plants. It was presumed to be a member of narnaviruses. MaNLV has the same characteristics as narnaviruses, and its genomic sequence only encodes a conserved RdRp protein. The extent of the virus infection and whether it causes symptoms need to be confirmed by subsequent experiments (Appendix A).

The International Committee on Taxonomy of Viruses recognizes the Citrus leaf blotch virus (CLBV) as the sole species in the genus *Citrivirus* [23]. The natural host range of CLBV is citrus, and it has also been reported in *Prunus* spp. [28], Actinidia [29], peony [30], and mulberry [16]. Studies have shown that for juvenile citrus plants, an advantage of CLBV as a viral vector for citrus is that it causes asymptomatic infection in most cultivars [31]. CLBV vectors are appropriate for gene expression or silencing in tissues that lack phloem, including meristematic regions [24], and CLBV is not transmitted by vectors. Therefore, they could be safely used in field experiments [32]. In our study, we identified a new CLBV isolate from mulberry trees named CLBV-ML2. It is highly homologous with the whole sequence and ORF1 of the previously reported CLBV-ML [16] but differs from ORF2 and ORF3. Recently, a viral vector based on the Citrus leaf blotch virus for genetics and breeding, which can used for precocious flowering of juvenile citrus. The mulberry tree harboring CLBV-ML2 showed no symptoms at the date of collection. Therefore, it has a good application potential as a virus-mediated transgenic vector using mulberry as the host if the virus can confirm implement system infection. 

In the ensuing decades, researchers focused on discovering and eliminating viral threats to plant and animal health. However, viruses are not merely agents of destruction but essential components of global ecosystems. Due to the increasing interest in organic farming to reduce the use of synthetic chemicals in agriculture, viral-based vector to manage crop pests and diseases in a sustainable way is a good strategy. A virus-mediated transgenic system is a quick system to deliver genes without any need for transformation and regeneration techniques and has currently been successfully used in apples and other woody species [33,34,35,36,37,38,39,40,41,42]. More efforts need to be made to discover new viruses in mulberry to accelerate CLBV-ML2 the application of virus-mediated transgenic in mulberry varieties.

## 5. Conclusions

In this study, we collected eight groups of mulberry leaves from different provinces in China. Three viruses were identified based on genome and phylogenetic analysis, named Citrus leaf blotch virus isolate mulberry alba 2 (CLBV-ML2), Mulberry-associated virga-like virus (MaVLV), and Mulberry-associated narna-like virus (MaNLV).

## Figures and Tables

**Figure 1 viruses-14-02564-f001:**
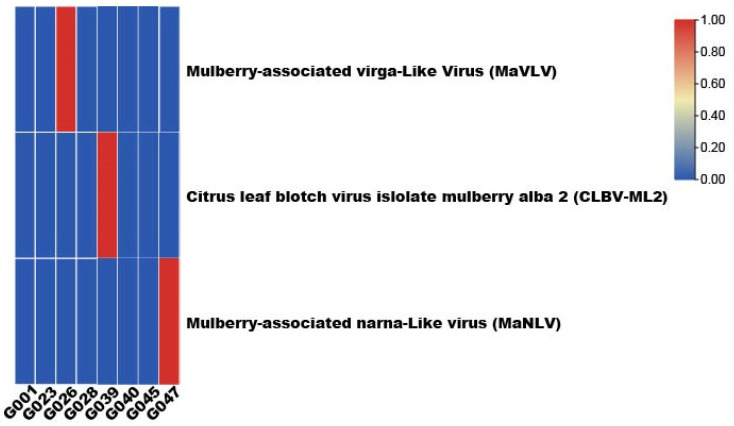
The heatmap shows the distribution of MaVLV, CLBV-ML2, and MaNLV in samples.

**Figure 2 viruses-14-02564-f002:**
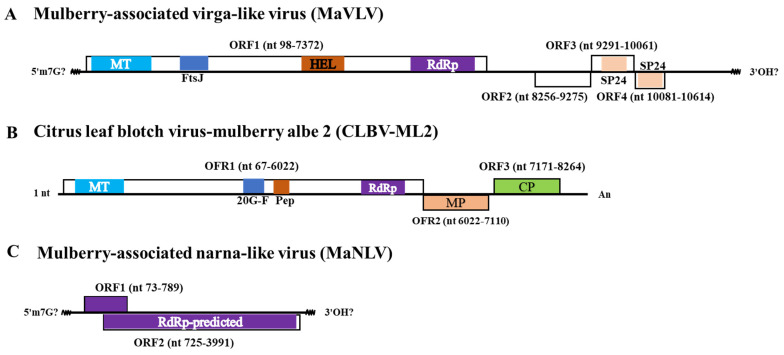
Predicted open reading frames (ORFs, with boxes), conserved motifs, domains, and viral proteins (different colors). (**A**) MaVLV, (**B**) CLBV-ML2, and (**C**) MaNTL. Viral methyltransferase (MT, pfam01660), viral helicase (HEL, pfam01443), and RNA-dependent RNA polymerase_2 (RdRp, pfam00978), 2OG-FeII Oxy superfamily (2OG-F, cl21496), peptidase C23 superfamily (Pep, cl05111), SP24 (pfam16504), coat protein (CP) and movement protein (MP). The unknown sequences of 5′ and 3′ UTRs are shown by black lines at both extremities, An: poly A tail.

**Figure 3 viruses-14-02564-f003:**
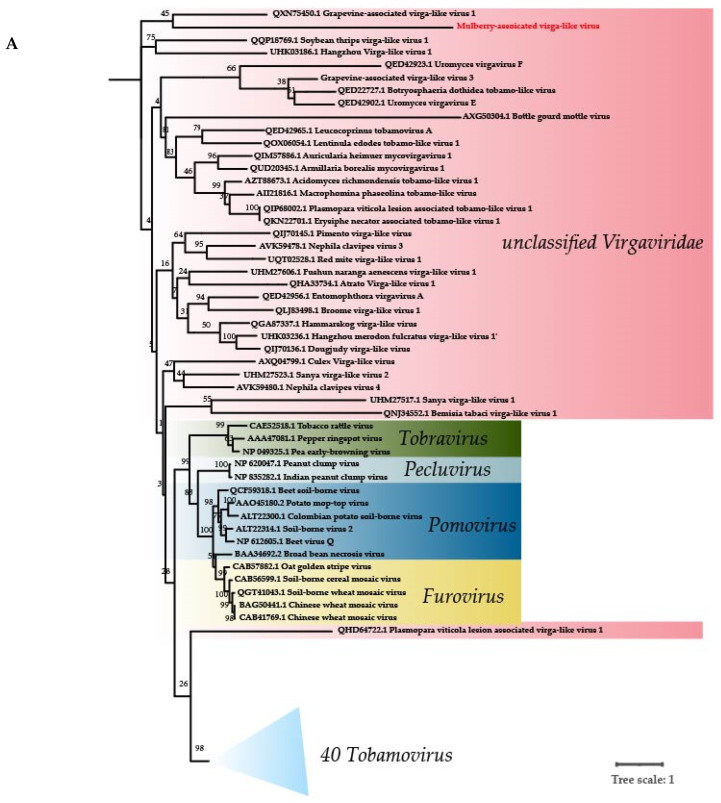
Phylogenetic trees of (**A**) RdRp proteins constructed for representative members of the family *Virgaviridae* (40 tobamoviruses; 5 furoviruses; 6 pomoviruses; 2 pecluviruses; 3 tobraviruses and 32 unclassified virgaviridaes). (**B**) The pairwise identity plots of the RNA-dependent RNA polymerase amino acid sequences aligned by ClustalW and displayed by sequence demarcation using TBtools software. The trees were constructed using the maximum likelihood method, and the statistical significance of the branches was evaluated by bootstrap analysis (1000 replicates). Newly discovered viruses are marked in red. 40 tobamovirus are collapsed together. Relationship: pink background.

**Figure 4 viruses-14-02564-f004:**
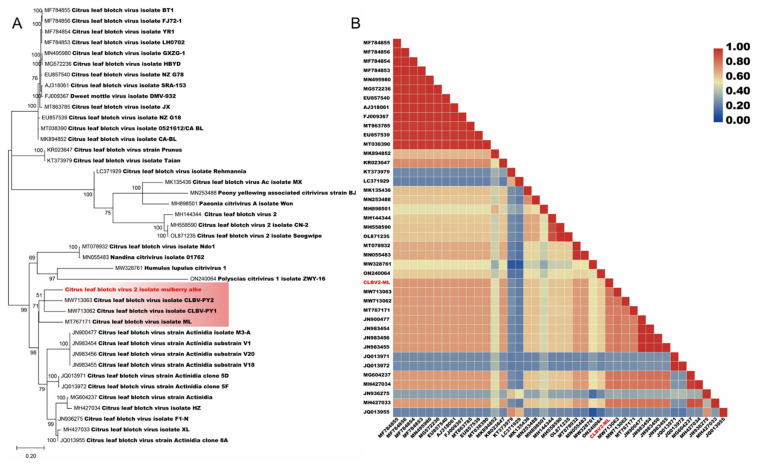
Phylogenetic trees constructed for representative members of the *Citrivirus* genus. (**A**) The whole sequences for phylogenetic trees. (**B**) The pairwise identity plots of the complete genomic nucleotide sequences aligned by ClustalW and displayed by sequence demarcation using TBtools software. The trees were constructed using the maximum likelihood method, and the statistical significance of the branches was evaluated by bootstrap analysis (1000 replicates). Newly discovered viruses are marked in red; relationship: pink background.

**Figure 5 viruses-14-02564-f005:**
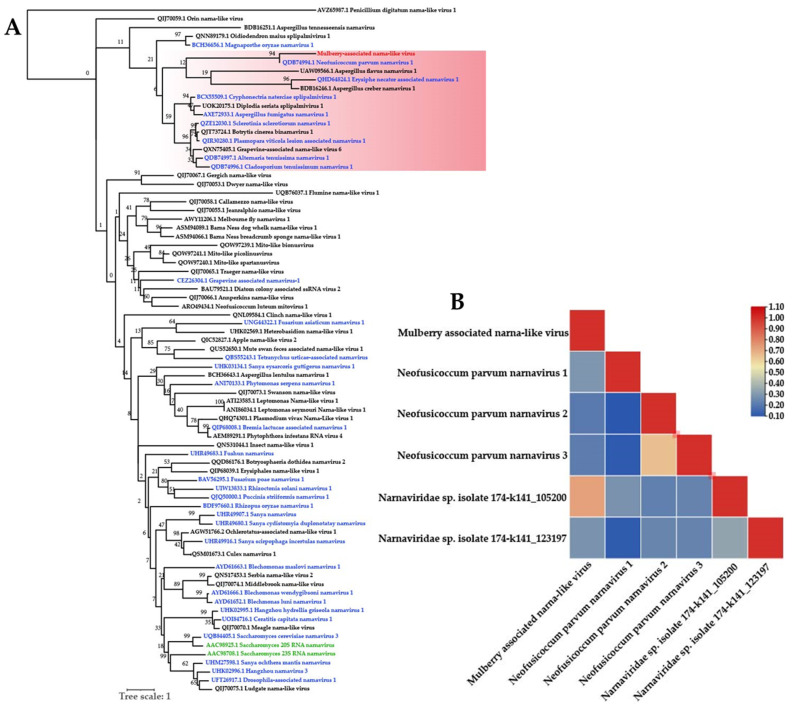
Phylogenetic trees of (**A**) RdRp proteins constructed for representative members of the family *Narnaviridae* (43 unclassified narnaviridaes; 32 unclassified narnaviruses; 2 narnaviruses). (**B**) The pairwise identity plots of the RNA-dependent RNA polymerase amino acid sequences aligned by ClustalW and displayed by sequence demarcation using TBtools software. The trees were constructed using the maximum likelihood method, and the statistical significance of the branches was evaluated by bootstrap analysis (1000 replicates). Newly discovered viruses are marked in red, unclassified narnaviridaes in black, unclassified narnaviruses in bule, and narnaviruses in green. Relationship: pink background.

**Figure 6 viruses-14-02564-f006:**
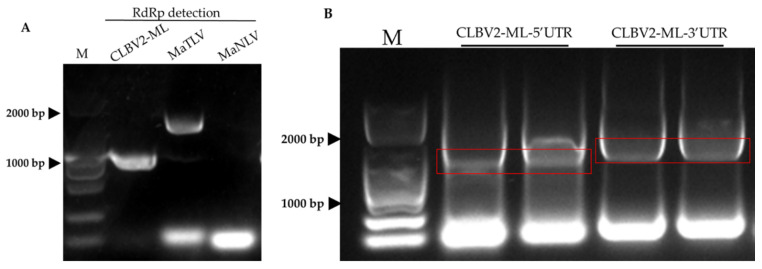
The detection of RdRp protein and UTR sequences. (**A**) The expression of RdRp proteins of CLBV-ML2, MaVLV, and MaNLV by RT-PCR. (**B**) RACE amplification of CLBV-ML2. M: Marker. Red boxes: 5’UTR and 3’UTR target strips.

**Figure 7 viruses-14-02564-f007:**
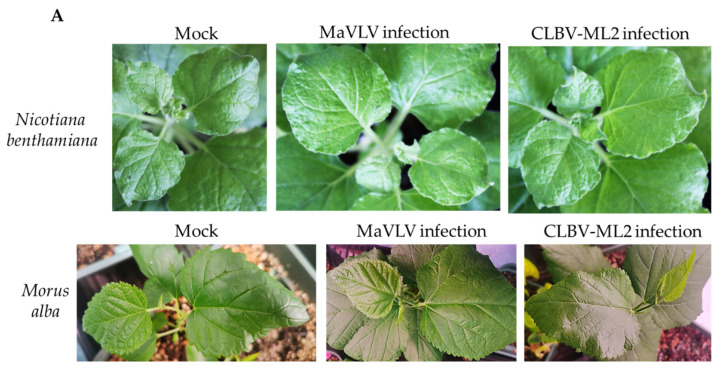
Symptoms on leaves of *N. benthamiana* and *Morus alba* infected by MaVLV, CLBV-ML2, and virus detection. (**A**) No symptoms on leaves of *N. benthamiana* and *Morus alba*. (**B**) RT-PCR detection of the infected leaves at 10 dpi. Lane 1 and 2: *N. benthamiana*, Lane 3: *Morus alba*, S: sample cDNA.

**Table 1 viruses-14-02564-t001:** Mulberry leaf sample information.

Number	Name	Province	Years	Latitude and Longitude	CLBV-ML2	MaVLV	MaNLV
G001	G-HEBBS2H	Heilongjiang	600	126°54′ N 45°80′ E	-	-	-
G023	G-QXGGZXZ	Henan	500	114°68′ N 34°58′ E	-	-	-
G026	G-NYXYSYZ	Henan	1800	112°46′ N 32°61′ E	-	+	-
G028	G-XJTL	Shandong	1200	116°01′ N 36°98′ E	-	-	-
G039	G-HSSX	Anhui	800	118°42′ N 29°86′ E	+	-	-
G040	G-BJDX1	Beijing	145	116°34′ N 39°73′ E	-	-	-
G045	G-NBQZ3	Xinjiang	425	121°55′ N 29°82′ E	-	-	-
G047	G-SXXC2	Xizang	1600	120°91′ N 29°50′ E	-	-	+

**Table 2 viruses-14-02564-t002:** Sequence similarities of CLBV-ML2 compared to Citrus leaf blotch virus (CLBV) isolates and unassigned citriviruses.

Name *	Genome	ORF1 (Rep)	ORF2 (MP)	ORF3 (CP)
CLBV-PY2	83.08/8792 nt	82.44 (88.11)	82.72 (95.58)	84.11 (94.21)
Dweet mottle virus	80.33/8747 nt	79.23 (78.63)	n.a. (97.79)	84.11 (94.77)
CLBV Strain Actinidia	81.95/8793 nt	81.07 (88.81)	n.a. (94.75)	84.59 (95.04)
CLBV	78.19/8747 nt	75.97 (78.00)	n.a. (98.34)	84.38 (95.32)
CLBV isolate ML	81.87/8776 nt	81.07 (87.29)	n.a. (95.58)	n.a. (88.43)
CLBV2	78.82/8697 nt	n.a. (58.07)	n.a. (90.61)	n.a. (93.66)
Humulus lupulus citrivirus 1	76.637181 nt	75.44 (63.95)	79.76 (86.19)	n.a. (90.61)
CLBV strain Prunus	78.17/8762 nt	n.a. (76.52)	n.a. (95.86)	n.a. (87.88)
Paeonia citrivirus A	79.28/6451 nt	n.a. (59.77)	84.02 (98.07)	83.12 (95.27)
Peony yellowing-associated Citrivirus	80.28/8666 nt	n.a. (53.35)	n.a. (96.13)	n.a. (88.71)
Nandina Citrivirus	77.74/8892 nt	77.66 (81.18)	n.a. (58.07)	n.a. (78.51)

* The GenBank accession numbers used are as follows: CLBV-PY2: MW713063, Dweet mottle virus: FJ009367, CLBV strain Actinidia: MG604237, CLBV: MN495980, CLBV isolate ML: MT767171, CLBV2: MH144344, Humulus lupulus citrivirus 1: MW328761, CLBV strain Prunus: KR023647, Paeonia citrivirus A: MH898501, Peony yellowing-associated Citrivirus: MN253488 and Nandina Citrivirus: MN055483. Comparison based on amino acid sequence is shown in parentheses.

## Data Availability

Appendix A are provided. Raw data are available upon request. The raw sequence data reported in this paper have been deposited in the Genome Sequence Archive in BIG Data Center, Beijing Institute of Genomics (BIG), Chinese Academy of Sciences (https://ngdc.cncb.ac.cn/, accessed on 15 January 2022), under accession numbers PRJCA013141.

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
