# Peer review of "Identification of Three Viruses Infecting Mulberry Varieties"

_viruses, 2022, doi:10.3390/v14112564_

Round 1
Reviewer 1 Report
In the manuscript of Chen et al three viruses found on mulberry in China have been characterized. Although there is little news in the article, it could be published, if the virus genomes and their taxonomic position were more fully characterized.
Main comments:
The approximate age of these trees should be indicated in the Materials and Methods section, and the word “ancient” should be then deleted throughout of the manuscript.
The affiliation of MaTLV to the genus Tobamovirus has not been proven. Firstly, the genome of a typical tobamovirus is much shorter (usually about 6.5 kb). Secondly, on the phylogenetic tree (Fig. 3A), the MaTLV isolate is clearly out of the tobamovirus cluster. Thirdly, the closest relatives of MaTLV (HVLV18, BVLV1) are unclassified virgaviruses found in invertebrates. Apparently, MaTLV is not tobamovirus. The authors should clarify the taxonomic position of MaTLV. Sequence identity between tobamoviruses and MaTLV should also be calculated and demarcation criteria for the genus Tobamovirus should be indicated. It seems to me that analyzing and discussing a possible connection between MaTLV and its closest relatives from invertebrate hosts is very important and needs a more detailed discussion.
Citrus leaf blotch virus was previously detected on mulberry (MT767171). Even though the full-length genomes of CLBV-ML and CLBV2-ML shares 81,87% identity, the demarcation criteria for the genus Citrivirus are not indicated. Without it, the isolate CLBV2-ML cannot be considered a novel virus of mulberry. Moreover, the authors themselves point out that the only difference between the known and the virus isolate found in this work is that CLBV-ML causes symptoms on mulberry, while CLBV2-ML does not.
MaNLV virus was most closely related (79.21%) to some representative of the family Narnaviridae (MZ680060), which was found in pond sediment. It is quite unclear why the author considere MaNLV as a fungal virus.
Alfalfa mosaic virus and cowpea mosaic virus are not tobamoviruses. They belong to the genus Alfalfavirus and Comovirus, respectively.
According to the Instructions for Authors, “new sequence information must be deposited to the appropriate database prior to submission of the manuscript. Accession numbers provided by the database should be included in the submitted manuscript”. However, I didn't find any mention in the manuscript that genome sequences of the three viruses detected were deposited in GenBank and Sequence Read Archive.
In the Reference section the author names should be presented in full.
Because it was very inconvenient to review the manuscript without line and page numbering, some other comments are inserted in the pdf version of the manuscript.
Two general comments:
The assumption about the possible benefits of the three detected viruses for the infected mulberry plants is not based on anything. There are no relevant experiments in the article. The fact that viruses are latent does not mean that they are beneficial. Therefore, all discussions of this issue should be removed throughout the manuscript.
A large place in the discussion section is occupied by arguments about the possibility of using the found viruses for the construction of transgenic plants. It may be so, but since the authors do not report any achievements in this direction, it would be possible to be limited to one sentence or a short paragraph.
Given the large number of factual errors, I believe that the manuscript cannot be published in its present form and should be largely rewritten.

Author Response
Reviewer #1: The approximate age of these trees should be indicated in the Materials and Methods section, and the word “ancient” should be then deleted throughout of the manuscript.
We delete the word “ancient” throughout of the manuscript, and add the age of these trees in the table 1 in the Materials and Methods section.
Reviewer #2: The affiliation of MaTLV to the genus Tobamovirus has not been proven. Firstly, the genome of a typical tobamovirus is much shorter (usually about 6.5 kb). Secondly, on the phylogenetic tree (Fig. 3A), the MaTLV isolate is clearly out of the tobamovirus cluster. Thirdly, the closest relatives of MaTLV (HVLV18, BVLV1) are unclassified virgaviruses found in invertebrates. Apparently, MaTLV is not tobamovirus. The authors should clarify the taxonomic position of MaTLV. Sequence identity between tobamoviruses and MaTLV should also be calculated and demarcation criteria for the genus Tobamovirus should be indicated. It seems to me that analyzing and discussing a possible connection between MaTLV and its closest relatives from invertebrate hosts is very important and needs a more detailed discussion.
Based on the comments from reviewer, we realized that it might not be appropriate to use the full length of the viruses sequences for phylogenetic tree creation. So the new phylogenetic tree was analysed based on the RNA-dependent RNA polymerase (RdRp) amino acid sequences of representative members of the Virgaviridae family (Including unclassified members, Figure 3A). According to the new results, of which MaTLV falls in the clade with Grapevine-associated virga-like virus (QXN75450), we changed its name to Mulberry-associated virga-like virus (MaVLV). In the correlation heatmap of RdRp sequences, MaVLV showed higher similarity with the isolate of GaVLV than HVLV18 and BVLV1. We changed the conclusion and proposed that MaVLV was the member of the unclassified Virgaviridae. The evidence suggests that: MaVLV ORF1 is an alpha-like replication protein with conserved methyltransferase (MT) and helicase (Hel) domains, and the RNA-dependent RNA polymerase (RdRp) is expressed as the C-terminal part of this protein by readthrough of a leaky stop codon, which was similar to the member of virgaviridae (1,2).
Reviewer #3: Citrus leaf blotch virus was previously detected on mulberry (MT767171). Even though the full-length genomes of CLBV-ML and CLBV2-ML shares 81,87% identity, the demarcation criteria for the genus Citrivirus are not indicated. Without it, the isolate CLBV2-ML cannot be considered a novel virus of mulberry. Moreover, the authors themselves point out that the only difference between the known and the virus isolate found in this work is that CLBV-ML causes symptoms on mulberry, while CLBV2-ML does not.
We propose that CLBV-ML2 is a new isolate of Citrivirus based on the species demarcation criteria for Betafexiviridae (3), we change the name to Citrus leaf blotch virus-mulberry alba 2 (CLBV-ML2). The nucleotide identity among CLBV-ML2, CLBV isolates and unassigned citriviruses are higher than 75 % (Table 2). The whole sequence and ORF1 of CLBV-ML2 are highly homolgous with CLBV-ML , but there is very low homology in ORF2 and ORF3.
Reviewer #4: MaNLV virus was most closely related (79.21%) to some representative of the family Narnaviridae (MZ680060), which was found in pond sediment. It is quite unclear why the author considere MaNLV as a fungal virus.
Based on the comment from Reviewer, we realize that the definition of MaNLV as a fungal virus is not appropriate. After going through more research paper, we know that narnaviruses could be detected not only in diverse fungi, but also plants, protists, arthropods, and nematodes. And they could encodeonly a highly conserved RNA-dependent RNA polymerase (RdRp)(4). The new phylogenetic tree based on the RNA-dependent RNA polymerase (RdRp) amino acid sequences of representative members of the Narnaviridae family (Including unclassified members, Figure 5A). MaNLV falls in the clade with Neofusicoccum parvum narnavirus, which was found in grapevine wood tissues and a member of unclassified narnaviruses. MaNLV has the same characteristics with narnaviruses, and its genomic sequence only encodes a conserved RdRp protein. we considered that MaNLV belongs to the genus Narnavirus.
Reviewer #5: According to the Instructions for Authors, “new sequence information must be deposited to the appropriate database prior to submission of the manuscript. Accession numbers provided by the database should be included in the submitted manuscript”. However, I didn't find any mention in the manuscript that genome sequences of the three viruses detected were deposited in GenBank and Sequence Read Archive.
We have deposited the genome sequences of the three detected viruses into the National Genomics Data Center. The data under review.
Reviewer #6: In the Reference section the author names should be presented in full.
As suggested, we revised it accordingly.
Reviewer #7: Because it was very inconvenient to review the manuscript without line and page numbering, some other comments are inserted in the pdf version of the manuscript.
As suggested, we revised it accordingly.
Reviewer #8: The assumption about the possible benefits of the three detected viruses for the infected mulberry plants is not based on anything. There are no relevant experiments in the article. The fact that viruses are latent does not mean that they are beneficial. Therefore, all discussions of this issue should be removed throughout the manuscript.
As suggested, we revised it, we deleted the part of the speculate about possible benefits of the three detected viruses for the infected mulberry plants.
Reviewer #9: A large place in the discussion section is occupied by arguments about the possibility of using the found viruses for the construction of transgenic plants. It may be so, but since the authors do not report any achievements in this direction, it would be possible to be limited to one sentence or a short paragraph.
As suggested, we revised it. limited to a short paragraph in discussion

Reviewer 2 Report
The article Chen et al investigates and identifies novel viruses infecting Mulberry plants. Overall the study is important for readers however, there is scope to improve the manuscript before publishing.
Major critical comments:
- line numbers are missing which makes it difficult for a reviewer to give a specific comment and would request authors to add it for peer-review.
- Some of the citations are not relevant to the context. e.g., viruses enhance hosts' ability to abiotic stresses should cite the pioneer study from Roossinck lab that showed viral infection makes plant drought tolerant.
- As this is an original article, the introduction and discussion should be much more elaborate to highlight the broad scope of this study. In the second paragraph of Introduction, authors talked about ancient mulberries being good for identifying novel beneficial viruses and failed to cite relevant papers. Overall, it should be also highlighted here how transcriptome sequencing is helping in identifying many novel viruses from woody plants which otherwise will be difficult to detect because of either low viral titer or difficulty of RNA extractions from woody plants or the absence of any symptoms. Recently, transcriptome sequencing was used in citrus to identify a novel latent viral infection and analysis showed that they can be ancestral viruses to the current umbravirus. https://doi.org/10.3390/v13040646; https://doi.org/10.3389/fmicb.2021.683130.
- The references are not properly cited either within the text or at the end of the manuscript. I would request authors to follow the latest articles from the journal Viruses to add references in the correct format.
- MaTLV: some time this has been referred as Mulberry-associated Tobamo-llike virus and Mulberry-associated Tobacco-like virus. Please be consistent with the naming.
- methodology 2.5: Please provide the full protocol of RT PCR. what PCR conditions were used. In Figure 6, I would request to provide better pictures where the bands are clearly visible. In the legend of figure 6, it should be lane and not line. section 3.4, does not explain figure 6 properly, it should be written here how many plants were inoculated and how many plants got infected. When a new virus is identified normally it is inoculated in at least 10 plants or more for proper characterization.
- section 3.2: "Virgaviridae family shows that MaTLV identified in the present study falls in the outgroup where sequences of Tobamovirus" - outgroup is the not the correct term here. Outgroup is used to indicate the rooting of the tree. Here, it can be said that MaTLV makes a strong clade with Tobamovirus suggesting a similar sequence to tobamoviruses. Moreover, in the legend of the phylogenetic trees, it should explain what is the root? mid-point root or rooted to an outgroup.
- In the discussion, authors can also discuss organic farming which is nowadays a very active research field and virus-mediated transgenic vector can replace harmful synthetic chemicals to manage crop diseases in a sustainable way. https://doi.org/10.1007/s11356-021-15258-7
Author Response
Reviewer #1: line numbers are missing which makes it difficult for a reviewer to give a specific comment and would request authors to add it for peer-review.
As suggested, we revised it accordingly.
Reviewer #2: Some of the citations are not relevant to the context. e.g., viruses enhance hosts' ability to abiotic stresses should cite the pioneer study from Roossinck lab that showed viral infection makes plant drought tolerant.
According to other reviewer’s suggestion, we have deleted this sentence in the manuscript.
Reviewer #3: As this is an original article, the introduction and discussion should be much more elaborate to highlight the broad scope of this study. In the second paragraph of Introduction, authors talked about ancient mulberries being good for identifying novel beneficial viruses and failed to cite relevant papers. Overall, it should be also highlighted here how transcriptome sequencing is helping in identifying many novel viruses from woody plants which otherwise will be difficult to detect because of either low viral titer or difficulty of RNA extractions from woody plants or the absence of any symptoms. Recently, transcriptome sequencing was used in citrus to identify a novel latent viral infection and analysis showed that they can be ancestral viruses to the current umbravirus. https://doi.org/10.3390/v13040646;
https://doi.org/10.3389/fmicb.2021.683130.
Thanks a lot for these great suggestions. As suggested, we revised it in the introduction. Two references provided by the reviewer were used to prove that the rapid development of NGS brought about the convenience of identifying new viruses in woody plants
Reviewer #4: The references are not properly cited either within the text or at the end of the manuscript. I would request authors to follow the latest articles from the journal Viruses to add references in the correct format.
We revised it accordingly.
Reviewer #5: MaTLV: sometime this has been referred as Mulberry-associated Tobamo-llike virus and Mulberry-associated Tobacco-like virus. Please be consistent with the naming.
Based on the new evidence from phylogenetic tree and sequence correlation, we change the Mulberry-associated Tobamo-like (MaTLV) to Mulberry-associated virga-like (MaVLV), and we propose that MaVLV was the member of the unclassified Virgaviridae.
Reviewer #6: methodology 2.5: Please provide the full protocol of RT PCR. what PCR conditions were used. In Figure 6, I would request to provide better pictures where the bands are clearly visible. In the legend of figure 6, it should be lane and not line. section 3.4, does not explain figure 6 properly, it should be written here how many plants were inoculated and how many plants got infected. When a new virus is identified normally it is inoculated in at least 10 plants or more for proper characterization.
As suggested, we revised it in materials and methods, mechanical inoculation and virus detection by RT-PCR. The picture in Figure 7 (Figure 6 before) have been replaced.
Reviewer #7: section 3.2: "Virgaviridae family shows that MaTLV identified in the present study falls in the outgroup where sequences of Tobamovirus" - outgroup is the not the correct term here. Outgroup is used to indicate the rooting of the tree. Here, it can be said that MaTLV makes a strong clade with Tobamovirus suggesting a similar sequence to tobamoviruses. Moreover, in the legend of the phylogenetic trees, it should explain what is the root? mid-point root or rooted to an outgroup.
Thank you for your comments, we realized that there were errors in the understanding of outgroup in the manuscript. Actually, we didn’t use an outgroup to indicate the rooting of the tree in phylogenetic tree. So we deleted the part.
Reviewer #8: In the discussion, authors can also discuss organic farming which is nowadays a very active research field and virus-mediated transgenic vector can replace harmful synthetic chemicals to manage crop diseases in a sustainable way. https://doi.org/10.1007/s11356-021-15258-7
We revised it accordingly. And cited the reference provided by the reviewer

Reviewer 3 Report
The authors collected eight mulberry samples from seven areas, three of which were identified to carry viral sequences based on RNA sequencing analysis. The full-length genome of CLBV2-ML was obtained after 5' and 3' RACE was performed, while sequences of MaTLV and MaNLV are still incomplete. Mulberry was identified as a new host of MaTLV and MaNLV. Sequence and phylogenetic analysis were conducted to show the correlation of viral sequences obtained in this study with that downloaded from NCBI. Infectivity of three viruses were tested by sap inoculation on Nicotiana benthamiana and Morus alba, and the authors found CLBV2-ML could infect the seedlings of both host plants, while MaTLV could only infect N. benthamiana. However, MaNLV failed to infect both plants.
This study was to identify novel viruses from ancient mulberry plants using next-generation technique and the authors proposed the application potential of these viruses in developing transgenic mulberry variety with good characteristics. The aim of this study is good; however, many contents and results must be improved.
Major issue:
1. The title of this manuscript is inappropriate. These three viruses are not new viruses. Moreover, mulberry plant has been determined as a host of CLBV2 by the other group.
2. As for the abstract part, the authors mentioned poor knowledge of beneficial viruses which might enhance their host adaptation to different biotic and abiotic stress. I expected the authors would perform associated experiments to prove the viral strain they identified have such potential. Otherwise, there is no need to mention about it.
3. More information about the occurrence or viral diseases on mulberry must be included in introduction part and point out the novelty point of this study.
4. Virus challenge on two different plants showed no symptom on the top and RT-PCR bands are quite weak. One possibility is viral titers in collected mulberry was low. The suggestion for that is to use concentrated sap. The other possibility is inoculation buffer is inappropriate, especially pH.
The increase of virus replication on inoculated and systemic leaves should be monitored at different time points to show the infectivity, which can also exclude sap contamination issue. Furthermore, construction of cDNA infectious clone should be considered.
If virus cannot replicate or move efficiently in host or virus infection is completely suppressed by host immune system, I don’t see the application potential of such virus. At least it can systemically infect plants. However, only inoculated leaves were detected in this study.
5. The authors pointed out that MaNLV is a fungal virus. So the mRNA info of host fungus should be found in RNA seq data. If not, this MaNLV could infect plants by their own. If yes, please clarify it.
6. Data amount of this study is not enough. Maybe more samples should be included in this study.
7. The authors must re-edit each sentence very carefully. More than 4 comma are even found in one single sentence. Sentences are suggested to be simplified. Upper case letters showed in the middle of some sentences. Grammatical issues throughout the manuscript must be corrected. Some mistakes are shown in Minor issues.
Minor issue:
1. Inappropriate use of English words.
a) “Threat” is a noun and “threaten” is a verb. At Page 1
b) “Purification of PCR product was cloned….” At Page 3. Corrected as “purified PCR product”.
c) “can be detection” at Page 7. Corrected as “can be detected”
d) “the 66nt 5’UTR and 528 nt 3’UTR were found” at Page 7. Corrected as”…were obtained”
e) “insufficient toxic sources” at Page 8. Corrected as “low viral titer”
f) “Only viruses that have no adverse effects on the host…” at Page 1. Corrected as “minor adverse effect on the host…”
g) “do harmful to” at Page 1. Corrected as “do harm to…”
2. A Genbank accession number is required for at least CLBV2-ML.
3. “The present study examined 8 different species of ancient mulberry” at Page 2. If they are 8 different species, please clarify their latin name in Table 1.
4. The format of references is not unified.

Author Response
Reviewer #1: The title of this manuscript is inappropriate. These three viruses are not new viruses. Moreover, mulberry plant has been determined as a host of CLBV2 by the other group.
As suggested, we revised it, and change the title to “Identification of Three Viruses Infecting Mulberry Varieties”
Reviewer #2: As for the abstract part, the authors mentioned poor knowledge of beneficial viruses which might enhance their host adaptation to different biotic and abiotic stress. I expected the authors would perform associated experiments to prove the viral strain they identified have such potential. Otherwise, there is no need to mention about it.
Thanks a lot for this great suggestion. As suggested, we revised it, and deleted that part.
Reviewer #3: More information about the occurrence or viral diseases on mulberry must be included in introduction part and point out the novelty point of this study.
As suggested, we supplemented the viruses identified in mulberry trees in recent years, which would seriously affect their quality and yield. However, many viruses were still not identified, especially those with low virus concentration or weak symptoms. Therefore, we used NGS to explore these viruses from old mulberry trees lived for hundreds or thousands of years.
Reviewer #4: Virus challenge on two different plants showed no symptom on the top and RT-PCR bands are quite weak. One possibility is viral titers in collected mulberry was low. The suggestion for that is to use concentrated sap. The other possibility is inoculation buffer is inappropriate, especially pH. The increase of virus replication on inoculated and systemic leaves should be monitored at different time points to show the infectivity, which can also exclude sap contamination issue. Furthermore, construction of cDNA infectious clone should be considered. If virus cannot replicate or move efficiently in host or virus infection is completely suppressed by host immune system, I don’t see the application potential of such virus. At least it can systemically infect plants. However, only inoculated leaves were detected in this study.
Thanks for your suggestions, we will improve the way we inoculate the virus and build an infectious clone.
Reviewer #5: The authors pointed out that MaNLV is a fungal virus. So the mRNA info of host fungus should be found in RNA seq data. If not, this MaNLV could infect plants by their own. If yes, please clarify it.
As suggested, we realize that the definition of MaNLV as a fungal virus is not very strict. After going through more research paper, we know that narnaviruses could be detected not only in diverse fungi, but also plants, protists, arthropods, and nematodes.. And they could encodeonly a highly conserved RNA-dependent RNA polymerase (RdRp)(4). The new phylogenetic tree based on the RNA-dependent RNA polymerase (RdRp) amino acid sequences of representative members of the Narnaviridae family (Including unclassified members, Figure 5A). MaNLV falls in the clade with Neofusicoccum parvum narnavirus, which was found in grapevine wood tissues and a member of unclassified narnaviruses. MaNLV has the same characteristics as narnaviruses, and its genomic sequence only encodes a conserved RdRp protein. we considered that MaNLV belongs to the genus Narnavirus.
Reviewer #6: Data amount of this study is not enough. Maybe more samples should be included in this study.
Samples in this study were collected from mulberry plants lived for hundreds or thousands of years from different provinces in China. So it is unlikely to increase the number of samples.
Reviewer #7: The authors must re-edit each sentence very carefully. More than 4 commas are even found in one single sentence. Sentences are suggested to be simplified. Upper case letters showed in the middle of some sentences. Grammatical issues throughout the manuscript must be corrected. Some mistakes are shown in Minor issues.
As suggested, we revised a lot of statements.

Round 2
Reviewer 1 Report
The authors have addressed my comments. I have no objection to the publication of the manuscript
Author Response
Comment 1:
The authors have addressed my comments. I have no objection to the publication of the manuscript
Response:
Thank you for your kind comments. We added data availability at the end of the manuscript: The raw sequence data reported in this paper have been deposited in the Genome Sequence Archive in BIG Data Center, Beijing Institute of Genomics (BIG), Chinese Academy of Sciences (https://ngdc.cncb.ac.cn/), under accession numbers PRJCA013141.
Reviewer 2 Report
I thank the authors for improving the scientific rigor of the manuscript. The new gel figures shows clearly the infection in plants.
Some major issues still remain:
1- Figure 3: Phylogenetic tree for virga-like virus: the bootstrap values are not indicated while the legend says more than 50% are shown. Moreover, I would request the authors to improve all the phylogenetic trees. It is not critical to see all the viruses from Tombusvirus and thus, the nodes can be collapsed together. This helps to better visualise the clade where the virus of interest is present. The material and methods or the legend can indicate how many viruses from each genus were used to make the tree and make the current figure supplement figure.
lines 511 to 513: this can be edited for clarity: Due to the increasing interest in organic farming to reduce the use of synthetic chemicals in agriculture, viral based vector to manage crop pests and diseases in a sustainable way is a good strategy.
Author Response
Comment 1:
Figure 3: Phylogenetic tree for virga-like virus: the bootstrap values are not indicated while the legend says more than 50% are shown. Moreover, I would request the authors to improve all the phylogenetic trees. It is not critical to see all the viruses from Tombusvirus and thus, the nodes can be collapsed together. This helps to better visualise the clade where the virus of interest is present. The material and methods or the legend can indicate how many viruses from each genus were used to make the tree and make the current figure supplement figure.
Response:
As suggested, we revised it accordingly. First, we improve all the phylogenetic trees in the manuscript, add the bootstrap and collapse the unimportant groups. Second, we make the current figure supplement figure about how many viruses from each genus were used.
Comment 2:
lines 511 to 513: this can be edited for clarity: Due to the increasing interest in organic farming to reduce the use of synthetic chemicals in agriculture, viral based vector to manage crop pests and diseases in a sustainable way is a good strategy.
Response:
As suggested, we revised it accordingly.
Reviewer 3 Report
Major issues:
1. This manuscript still requires extensive editing of English language. A thorough proofreading is strongly suggested.
2. The method to identify new viruses in this study is not economic enough. It is unnecessary to prepare 8 different libraries for 8 samples. Only one library is required for mixed samples. To obtain more data, sequencing depth is also required. Then based on data mining and further analysis, different specific primers can be designed to check the origin of these viruses.
3. In the re-edited abstract, the authors mentioned that “there are still many viruses that have not been identified yet especially those either low accumulation or weak symptoms in mulberry trees.” From my perspective, the motivation of this study doesn’t have a profound effect on future research or application. My suggestion is that “Viruses-mediated genome editing in plants is a powerful strategy to develop plant cultivars with important and novel agricultural traits. Mulberry alba is an important economic tree species that has been cultivated in China for more than 5000 years. So far, only a few viruses have been identified from mulberry trees and their application potential is largely unknown. Therefore, mining more virus resources from mulberry tree can pave the way for the establishment of useful engineering tools. In this study…….”
4. The increase of virus replication on inoculated and systemic leaves should be monitored at different time points to show the infectivity, which can also exclude sap contamination issue. Otherwise, I don’t believe these viruses have infectivity.
Author Response
Comment 1:
This manuscript still requires extensive editing of English language. A thorough proofreading is strongly suggested.
Response:
As suggested, we revised it accordingly. We proofread and polished the manuscript thoroughly
Comment 2:
The method to identify new viruses in this study is not economic enough. It is unnecessary to prepare 8 different libraries for 8 samples. Only one library is required for mixed samples. To obtain more data, sequencing depth is also required. Then based on data mining and further analysis, different specific primers can be designed to check the origin of these viruses.
Response:
As suggested, we accept it and will apply it in the further research.
Comment 3:
In the re-edited abstract, the authors mentioned that “there are still many viruses that have not been identified yet especially those either low accumulation or weak symptoms in mulberry trees.” From my perspective, the motivation of this study doesn’t have a profound effect on future research or application. My suggestion is that “Viruses-mediated genome editing in plants is a powerful strategy to develop plant cultivars with important and novel agricultural traits. Mulberry alba is an important economic tree species that has been cultivated in China for more than 5000 years. So far, only a few viruses have been identified from mulberry trees and their application potential is largely unknown. Therefore, mining more virus resources from mulberry tree can pave the way for the establishment of useful engineering tools. In this study…….”
Response:
As suggested, we revised it accordingly. We have revised the summary of the manuscript.
Comment 4:
The increase of virus replication on inoculated and systemic leaves should be monitored at different time points to show the infectivity, which can also exclude sap contamination issue. Otherwise, I don’t believe these viruses have infectivity.
Response:
Thank you for your suggestion, which provides us with new knowledges. Due to the limited time for modification, we are sorry that we failed to improve the inoculation test. We will pay more attention to this in the future research.